# Identification of a Novel Oncogenic Fusion Gene *SPON1-TRIM29* in Clinical Ovarian Cancer That Promotes Cell and Tumor Growth and Enhances Chemoresistance in A2780 Cells

**DOI:** 10.3390/ijms23020689

**Published:** 2022-01-08

**Authors:** Saya Nagasawa, Kazuhiro Ikeda, Daisuke Shintani, Chiujung Yang, Satoru Takeda, Kosei Hasegawa, Kuniko Horie, Satoshi Inoue

**Affiliations:** 1Division of Systems Medicine & Gene Therapy, Saitama Medical University, Hidaka, Saitama 350-1241, Japan; sykondou@juntendo.ac.jp (S.N.); ikeda@saitama-med.ac.jp (K.I.); jiu-r@hotmail.com (C.Y.); 2Department of Obstetrics and Gynecology, Juntendo University School of Medicine, Bunkyo-ku, Tokyo 113-8421, Japan; stakeda@juntendo.ac.jp; 3Department of Gynecologic Oncology, Saitama Medical University International Medical Center, Hidaka, Saitama 350-1298, Japan; dshin@saitama-med.ac.jp (D.S.); koseih@saitama-med.ac.jp (K.H.); 4Department of Systems Aging Science and Medicine, Tokyo Metropolitan Institute of Gerontology, Itabashi-ku, Tokyo 173-0015, Japan

**Keywords:** fusion transcript, ovarian cancer, RNA-sequencing, small interfering RNA (siRNA), xenograft

## Abstract

Gene structure alterations, such as chromosomal rearrangements that develop fusion genes, often contribute to tumorigenesis. It has been shown that the fusion genes identified in public RNA-sequencing datasets are mainly derived from intrachromosomal rearrangements. In this study, we explored fusion transcripts in clinical ovarian cancer specimens based on our RNA-sequencing data. We successfully identified an in-frame fusion transcript *SPON1-TRIM29* in chromosome 11 from a recurrent tumor specimen of high-grade serous carcinoma (HGSC), which was not detected in the corresponding primary carcinoma, and validated the expression of the identical fusion transcript in another tumor from a distinct HGSC patient. Ovarian cancer A2780 cells stably expressing *SPON1-TRIM29* exhibited an increase in cell growth, whereas a decrease in apoptosis was observed, even in the presence of anticancer drugs. The siRNA-mediated silencing of *SPON1-TRIM29* fusion transcript substantially impaired the enhanced growth of A2780 cells expressing the chimeric gene treated with anticancer drugs. Moreover, a subcutaneous xenograft model using athymic mice indicated that *SPON1-TRIM29*-expressing A2780 cells rapidly generated tumors in vivo compared to control cells, whose growth was significantly repressed by the fusion-specific siRNA administration. Overall, the *SPON1-TRIM29* fusion gene could be involved in carcinogenesis and chemotherapy resistance in ovarian cancer, and offers potential use as a diagnostic and therapeutic target for the disease with the fusion transcript.

## 1. Introduction

Ovarian cancer is a common gynecological malignancy among females, which accounts for an estimated 313,959 new cases and 207,252 deaths worldwide per year [1]. Due to the small number of initial symptoms, the disease is often found at the advanced stage, based on the International Federation of Gynecology and Obstetrics (FIGO) staging system, and the 5 year survival rate of patients with obviously malignant tumors is ~50% [2]. Surgery and chemotherapy are the main therapeutic options for ovarian cancer; the latter is often used in both neoadjuvant and adjuvant settings in advanced-stage tumors. Ovarian cancer is subdivided into five histological types: high-grade serous carcinoma (HGSC), low-grade serous carcinoma, clear cell carcinoma (CCC), endometrioid carcinoma, and mucinous carcinoma [3]. HGSC is the most common histological subtype and is frequently treated with platinum/taxane-based chemotherapy using paclitaxel and carboplatin. The majority of HGSC tumors are initially sensitive to chemotherapy, although patients eventually suffer from tumor recurrence [2]. CCC and mucinous cancers are often chemotherapy-resistant, which results in poorer patient prognosis. Notably, the recent application of molecular targeted drugs in ovarian cancer treatment has improved patient prognosis. Among these drugs, poly (ADP-ribose) polymerase (PARP) inhibitor is a new option for maintenance therapy in patients with ovarian cancer who respond to platinum-based chemotherapy, particularly those who have tumors with the BRCA mutation or homologous recombination deficiency [4]. While the establishment of new therapeutic drugs has improved patient prognosis, therapy-refractory tumors still exist and remain to be treated. Therefore, it is important to understand the molecular mechanisms of therapy-refractory ovarian cancer for the development of alternative strategies.

The transcriptomic analysis of clinical tumor specimens is useful to understand gene expression profiles associated with the pathology. In particular, high-throughput RNA-sequencing (RNA-seq) reveals a comprehensive view of the entire transcriptome, including isoform-dependent expression, single-nucleotide polymorphisms, post-transcriptional modifications, and gene fusion [5]. We recently showed transcriptomic profiles of clinical ovarian cancers based on RNA-seq analysis and identified *CPNE8* and *BHLHE41* as prototypic upregulated genes, predominantly in CCC and HGSC, respectively [6]. Moreover, we identified new and known mutations of tumor-associated genes, such as *TP53*, in ovarian cancer specimens based on RNA-seq [7]. Our RNA-seq results also dissected the alterations in mutational status in some recurrent tumors compared with their matched-paired primary tumors. Thus, RNA-seq based studies, including ours, have provided new knowledge of the molecular mechanisms underlying ovarian cancer and will help to precisely understand the pathological characteristics of the disease [8].

Recent comprehensive studies of cancer genomes have further identified various chromosomal rearrangements, which often generate functional gene fusions encoding oncogenic proteins that can be applied to therapeutic targets, as exemplified by the *BCR-ABL* and *EML4-ALK* fusion genes in chronic myeloid leukemia [9] and non-small-cell lung cancer, respectively [10]. Pan-cancer studies in The Cancer Genome Atlas (TCGA) identified >25,000 fusion genes across 33 different tumor types [11,12,13]. In terms of ovarian cancer, a recent study of fusion predictions based on TCGA RNA-seq data revealed 9953 fusion predictions from 418 primary serous ovarian cancer tumors with a high observed rate of intrachromosomal fusions [14]. It is also notable that some of the fusion genes identified in ovarian cancer tumors were associated with the upregulation of neighboring genes, suggesting the remodeling of transcriptional regulation in the disease [14]. Nevertheless, therapeutic targetable fusion genes have not been identified in ovarian cancer.

In the present study, we performed RNA-seq analysis of 32 ovarian cancer tissues and 6 normal ovary tissues, and identified a novel fusion transcript *SPON1-TRIM29* from a recurrent HGSC tumor. Intriguingly, an identical fusion transcript was also expressed in another patient with HGSC. We showed a potential tumor-promotive function of *SPON1-TRIM29* based on gain- and loss-of-function studies in ovarian cancer cells. Our findings suggest that the *SPON1-TRIM29* fusion gene could contribute to tumor aggressiveness in a population of ovarian cancer and that the new biomarker may be applied as a therapeutic target for fusion-positive tumors.

## 2. Results

### 2.1. Identification of SPON1-TRIM29 In-Frame Fusion Transcript

A paired-end RNA seq was performed for 38 clinical ovary specimens, including 32 ovarian carcinomas and 6 normal ovary tissues (Table 1), based on the HiSeq 2500 platform (Illumina, San Diego, CA, USA). Among 32 ovarian carcinomas, 8 matched pairs of primary and recurrent tumors from 8 distinct patients were included. Our strategy for identifying fusion transcripts was to search for paired “chimeric” reads, with each read mapping to a different gene exon and fusing with each other in-frame (Figure 1a). Among the in-frame fusion predictions identified from 38 specimens, we selected candidate fusion transcripts that were expected to encode ≥50 amino acids for each gene with spanning reads ≥10 and spanning pairs ≥10 to minimize the cases of false positives. We identified a putative novel fusion transcript, *SPON1-TRIM29*, in chromosome 11 based on the criteria. *SPON1-TRIM29* was abundantly expressed in the recurrent HGSC specimen #S14R but not in its matched-pair primary tumor #S14 from an identical case. *SPON1* and *TRIM29* were located in chromosome 11 (11p15 and 11q23, respectively) and the putative fusion protein based on the RNA-seq data was estimated as the N-terminal signal peptide derived from *SPON1* (amino acids 1–115) and fused to a C-terminal half of *TRIM29* protein containing coiled-coil domain (amino acids 267–588) (Figure 1b). We designed RT-qPCR primers for the 3′-side of *SPON1* exon 2 and the 5′-side of *TRIM29* exon 2 to generate the amplified product, including the *SPON1-TRIM29* fusion point. Among 38 clinical ovary specimens, #S14R tumor was the only specimen that substantially expressed the *SPON1-TRIM29* fusion transcript (Figure 1c). Since the *SPON1-TRIM29* fusion transcript was only detected in S14R among RNA-sequenced samples, in RT-qPCR, the expression level of *SPON1-TRIM29* equivalent to S14R was considered as positive and the expression levels that were less than 1/10 of S14R were defined as negative. We also found that *TRIM29* expression was remarkably higher in #S14 and #S14R tumors compared with other HGSC tumors, as analyzed by RNA-seq (reads per kilobase of exon per million mapped sequence reads (rpkm) values: 109 and 554 for #S14 and #S14R, respectively; 15 ± 10 for a mean ± SD value among other HGSCs including #S1-#S13 and #S15), whereas *SPON1* expression was not elevated in #S14 and #S14R tumors compared with other HGSC tumors (rpkm values: 210 and 31 for #S14 and #S14R, respectively; 202 ± 212 for the mean ± SD value among other HGSCs).

As we discovered the *SPON1-TRIM29* in-frame fusion transcript in a HGSC tumor, we further investigated whether the fusion gene is expressed in other HGSC tumors. Among 100 specimens of HGSC clinical tumors from an independent cohort, we found that the substantial expression of *SPON1-TRIM29* fusion transcripts were observed in another patient, #O29. Sanger sequencing for the PCR-amplified fragments from #S14R and #O29 tumors revealed that the sequences of both fragments were identical, including the same fusion points (Figure 1d).

### 2.2. SPON1-TRIM29 Overexpression Stimulates Cell Proliferation

To analyze the function of SPON1-TRIM29, full-length flag-tagged *SPON1-TRIM29* cDNA was synthesized from #S14R and subcloned into pcDNA3 expression plasmid. The ectopic expression of SPON1-TRIM29 was confirmed by Western blot analysis using anti-Flag antibody in 293T cells transiently transfected with the *SPON1-TRIM29* expression vector (Appendix A). A2780 ovarian cancer cells stably expressing SPON1-TRIM29 or control vector were generated and the ectopic expression of SPON1-TRIM29 was validated by Western blot analysis in the fusion gene transfectants (SPON1-TRIM29-A2780 #A and #B) (Appendix A). We investigated the effects of SPON1-TRIM29 expression on cell proliferation using vector-transfected and *SPON1-TRIM29*-transfected A2780 cells with or without treatment with the platinum-based anticancer drug cisplatin, a taxan group chemical drug, paclitaxel, and a PARP inhibitor, olaparib (Figure 2). In the absence of chemotherapy drugs, *SPON1-TRIM29*-transfected cells exhibited a significant increase in cell proliferation compared with vector-transfected cells. Each chemotherapy drug generally impaired cell growth in a dose-dependent manner, although SPON1-TRIM29 expression significantly blunted the drug sensitivity in both low and high doses (Figure 2). In addition, a cell cycle analysis indicated that the cell population of G2/M phase decreased under cisplatin, paclitaxel, and olaparib treatments in SPON1-TRIM29-A2780 cells compared with Vector-A2780 cells (Appendix A). Since cisplatin, paclitaxel, and olaparib cause G2 arrest in cancer cells [15,16,17], SPON1-TRIM29 may be involved in the abrogation of these cancer drugs-induced G2 arrest.

### 2.3. SPON1-TRIM29 Inhibits Drug-Induced Apoptosis

We then examined the effect of SPON1-TRIM29 on apoptosis in A2780 cells by double-staining of annexin V and propidium iodide (PI). Since the difference in cell growth between the SPON1-TRIM29-A2780 and Vector-A2780 cells was most apparent at day 5 after the cancer drug administration, we examined apoptosis under the same conditions on day 5. Annexin V binds to cell surfaces expressing phosphatidylserine, an early apoptosis marker, and it is clear from previous studies that double staining with annexin V and PI distinguishes different stages of apoptotic cells, i.e., early (annexin V-positive/PI-negative) and late (annexin V-positive/P- positive) apoptotic cells [18]. In the present study, both early and late apoptotic cells were taken into account as apoptotic cells. Due to the culture time of 5 days, there were some PI-positive cells (10% or less) in the control groups (Figure 3a). Treatment with chemotherapy drugs, either cisplatin, paclitaxel, or olaparib, substantially increased annexin V-positive cell populations in both vector- and fusion-transfected cells compared with vehicle treatment (Figure 3a–d). Notably, the percentages of apoptotic cell populations based on annexin V staining were significantly lower in the *SPON1-TRIM29*-transfected cells compared with the vector-transfected cells (Figure 3e).

To investigate whether SPON1-TRIM29 expression substantially modulates cell proliferation and drug sensitivity in ovarian cancer cells, we designed siRNAs spanning the fusion sequence of *SPON1-TRIM29* (siFusion #A and #B) to specifically silence the *SPON1-TRIM29* fusion gene. Both siFusion #A and #B efficiently silenced the *SPON1-TRIM29* fusion transcripts in the SPON1-TRIM29-A2780 cells compared with the control siRNA (siLuc) (Appendix A). These siRNAs were transfected to the SPON1-TRIM29- and Vector-A2780 cells, and then cell a proliferation assay was performed in the presence or absence of drugs (Figure 4a–d). In the control siLuc transfection, SPON1-TRIM29-A2780 cells increased their growth compared with Vector-A2780 cells under the condition of vehicle and cancer drug treatment. When treated with siFusion, cell growth was repressed markedly in the SPON1-TRIM29-A2780 cells.

### 2.4. SPON1-TRIM29 Promotes In Vivo Growth of A2780 Cells

We further investigated whether SPON1-TRIM29 expression promotes in vivo tumor formation of ovarian cancer cells. We subcutaneously implanted vector-transfected or SPON1-TRIM29-expressing A2780 cells into female athymic mice. Nine days after the inoculation, the tumor volume in the mice bearing SPON1-TRIM29-A2780 cells was significantly larger compared with the mice bearing Vector-A2780 cells (Figure 5a,b). In addition, the tumor weights dissected from the SPON1-TRIM29-A2780 cell-bearing mice were significantly larger than those from the mice bearing Vector-A2780 cells (Figure 5c). Images of all the tumor-bearing mice and all the dissected tumors are presented in Appendix A, respectively.

Subsequently, we determined the effect of *SPON1-TRIM29* knockdown on SPON1-TRIM29-A2780 tumor growth in a xenograft model using athymic mice. SPON1-TRIM29-A2780 cells were implanted into the flanks of athymic female mice and were then injected with siFusion #1 or control siRNA (siControl). siFusion #1 injection significantly decreased the tumor volume and weight in the mice bearing SPON1-TRIM29-A2780 cells compared with siControl injection (Figure 5d–f). Images of all the tumor-bearing mice and all the dissected tumors are presented in Appendix A, respectively.

## 3. Discussion

In the present study, we identified a new fusion transcript *SPON1-TRIM29* in clinical ovarian cancer specimens based on RNA-seq analysis. Interestingly, the *SPON1-TRIM29* fusion was first detected in a recurrent HGSC tumor, whereas it was not detected in the corresponding primary cancer. Coincidentally, *SPON1-TRIM29* fusions were detected only in HGSC specimens from two distinct cases, implying that this transcript may play a role in the tumorigenesis of HGSC. Moreover, the overexpression and knockdown of *SPON1-TRIM29* demonstrated that this fusion gene product promotes tumor proliferation and chemoresistance in ovarian cancer cells. Notably, siFusion repressed cell growth markedly in SPON1-TRIM29-expressing cells, indicating that SPON1-TRIM29-expressing cells depend on the expression of SPON1-TRIM29. These results suggest that the *SPON1-TRIM29* fusion gene performs a tumor-progressive function.

Fusion genes have been considered to play a role in driving tumor initiation, metastasis, and resistance to therapy in solid cancers as well as in hematological malignancies [19,20]. A number of genes, including kinases, transcription factors, and cell cycle regulators, generate fusion genes in many types of cancers, and the resulting structural alteration and different regulation can cause tumorigenesis and progression. These fusion genes often contain kinase-fusion proteins, in which kinase activity is retained but out of regulation, leading to the constitutive activation of downstream genes responsible for tumor progression [21,22]. In various cancers, including glioblastoma, melanoma, breast, lung, and prostate, tyrosine kinases are the most representative of these kinases, e.g., ALK, ROS1, RET, and FGFR [23,24]. In clinic, ALK inhibitors are successfully used to treat non-small cell lung cancer, in which *ALK* fusion genes were found in approximately 5% [25]. In terms of ovarian cancer, a number of fusion genes has also been identified [26]; however, the elucidation of their functions and mechanisms is limited only in several fusion genes, including *SLC25A40-ABCB1*, *MAN2A1-FER*, *BCAM-AKT2*, *FHL2-GLI2*, *YWHAE-FAM22A*, *JAZF1-SUZ12*, and *FGFR3-TACC3* [27,28,29], and the clinical application of these findings is not available.

In terms of fusion genes in ovarian cancer, an integrated application with RNA-seq data also revealed *DPP9-PPP6R3* and *DPP9-PLIN3* fusion transcripts in HGSCs [30,31]. Dipeptidyl peptidase 9 (DPP9) is a member of the S9b protease family and *DPP9* mRNA expression is associated with longer survival for patients with uterine corpus endometrial carcinoma [32] and ovarian cancer [33]. Notably, these fusion genes feature the C-terminal-truncations of DPP9, putatively leading to a loss of its functional domains [30]. Another study based on TCGA RNA-seq data identified the *MUC1-TRIM46-KRTCAP2* fusion gene in HGSC [34]. The report shows that *MUC1-TRIM46-KRTCAP2* mRNA is not expressed in non-cancerous ovaries and its fusion protein localizes to cytoplasm without glycosylation in ovarian cancer cells, which is a similar feature of tumor-associated MUC1. MUC1 has been assumed to be a reliable diagnostic marker for ovarian cancer due to its overexpression in metastatic and advanced tumors [35,36]. Altogether, RNA-seq analysis is a useful platform that can dissect functional fusion transcripts associated with pathological features of malignancies, including ovarian cancer.

HGSC is known to feature mutations of tumor suppressor *TP53* at a high frequency [37]. Our previous report showed that the recurrent *SPON1-TRIM29*-positive specimen (#S14R) featured an R342X mutation in *TP53* mRNA, which was not found in the corresponding primary tumor specimen (#S14) [7]. R342X is a truncating mutation at codon 342 in the tetramerization domain of p53 protein, whose region is the most frequently mutated in various tumors, except its DNA binding domain [38,39]. The tetramerization domain is located in the C-terminus and its mutations often cause the non-functional or partial activity of p53 protein [40,41,42]. Thus, the resulting genomic instability may lead to the formation of the *SPON1-TRIM29* fusion gene, although the breakpoint is unknown. It is an important future issue to clarify the association between p53 mutation status and *SPON1-TRIM29* fusion, and their role in progression/anticancer drug resistance in ovarian cancer, especially based on histological types.

In line with the putative tumor-promotive function of SPON1-TRIM29 fusion protein, it may be relevant to dissect each function of SPON1 or TRIM29 in ovarian cancer. SPON1 is an extracellular matrix protein that is involved in neural cell adhesion and outgrowth [43], although its pathological role in cancers remains to be understood. While SPON1 expression is found in malignancies including ovarian cancer [44], the effect of SPON1 depends on the cancer type. In osteosarcoma cells, SPON1 promotes cell migration and invasion [45], whereas SPON1 exhibits an antioncogenic roles in some tumors: *SPON1* expression is negatively correlated with survival in bladder cancer [46], SPON1 mediates the apoptotic effect of anticancer reagent in uterine endometrial cancer cells [47], and SPON1 suppression leads to proliferation, migration, and invasion in hepatocellular carcinoma [48]. In the putative fusion protein based on the *SPON1-TRIM29* transcript, it contains 115 amino acids of SPON1 N-terminal signal sequence, whereas it lacks the rest of a large region containing reeler, spondin N, and thrombospondin type1 domains, which are conserved in spondin-like extracellular matrix proteins [49]. It is tempting to speculate that the truncated SPON1 protein of the SPON1-TRIM29 fusion affects proliferation/anticancer drug resistance in ovarian cancer cells in a dominant-negative manner.

TRIM29 is a member of the TRIM family of proteins, which harbors conserved domains of RING, B-box, and coiled-coil, although it does not contain the RING domain [50,51]. TRIM29 expression is associated with poor prognosis in various cancers, including cervical, pancreatic, lung, esophageal, and gastric cancers [52,53,54,55,56], but not in other cancers, such as squamous cell carcinoma [57]. In ovarian cancer, TRIM29 expression is associated with poor prognosis and is increased in the cisplatin-resistant ovarian cancer cells with increased cancer stem cell-like properties [58]. Previous cell experiment studies revealed the important roles of C- and N-terminal portions of TRIM29 proteins in cancers. TRIM29 protein interacts with DNA repair proteins through the C-terminal region (amino acids 394–588) to recruit them to chromatin in response to ionizing radiation, causing an increase in the radioresistance of cells [59]. TRIM29 also interacts with E3 ubiquitin ligase RNF8 through its C-terminal portion (amino acids 348–588), and promotes DNA damage response to ionizing radiation [60]. In addition, TRIM29 is phosphorylated by MAPKAP kinase 2 (MK2) at Ser550 in an ATM-dependent manner and performs a radioprotective function in pancreatic cancer cells [61]. Because the C-terminal region of TRIM29 (amino acids 267–588) is retained in the SPON1-TRIM29 fusion protein, it is possible that the molecule drives stress resistance in ovarian cancer cells in a dominant-positive manner. By contrast, the TRIM29 N-terminal region is absent in the SPON1-TRIM29 fusion protein; thus, the molecule does not exhibit some of the functions of TRIM29, such as inhibitory effects on NRF2 protein degradation by binding to KEAP1 [62], or on p53 activity by binding to p53 protein [63]. TRIM29 has been reported to interact with p53 through its N terminus and to inhibit p53 nuclear function by localizing p53 to the cytoskeleton [63]. Thus, it is possible that SPON1-TRIM29 modulates p53 activity through the perturbation of cytoskeletal sequestration. Further investigation of the mechanism and function of SPON1-TRIM29 fusion protein, including additional cell lines and primary cells, will provide useful information for the development of new therapeutic options for ovarian cancer with the aberrant molecule expression. Given that the ovarian cancer possesses *SPON1-TRIM29* gene fusion, treatment with SPON1-TRIM29 inhibitors, such as siRNA or small molecules, would increase the efficacy of cancer drugs in patients.

## 4. Materials and Methods

### 4.1. Clinical Specimens and Cell Lines

Experiments using patient data and specimens were approved by the Saitama Medical University International Medical Center Institutional Review Board (#13-165) [6,7]. In RNA-seq analysis, we used a total of 38 clinical specimens, including ovarian cancer and normal tissues, which were obtained from 30 distinct patients who underwent surgery for primary ovarian cancer with their informed consent in the previous study (#12-096): 8 paired ovarian cancer specimens of primary and recurrent tumors (5 serous, 1 clear cell, and 2 unclassified carcinomas), 10 serous carcinomas, 6 clear cell carcinomas, 5 normal ovarian tissues, and 1 normal oviduct tissue. In RT-qPCR analysis for the validation of fusion transcripts, we used 100 clinical HGSC specimens obtained from an independent cohort of 100 distinct patients. Human ovarian cancer A2780 cells and human kidney 293T cells were grown in RPMI1640 and DMEM, respectively, with 10% fetal bovine serum (FBS) and 1% penicillin/streptomycin at 37 °C under 5% CO_2_.

### 4.2. RNA-Seq

RNA was extracted from fresh frozen tissues using NucleoSpin RNA (Takara, Shiga, Japan). Quality control was performed via Agilent 2100 Bioanalyzer (Agilent, Palo Alto, CA, USA) and all RIN values were over 8.0. RNA-seq was performed as described previously [6]. RNA library was prepared by using SureSelect Strand Specific RNA Library Prep Kit (Agilent, Palo Alto, CA, USA) and 100 bp paired end RNA-seq was performed via HiSeq2500 (Illumina, San Diego, CA, USA). Ribosomal RNA and adopter sequence were deleted from FASTQ data. RNA-seq reads were aligned to the human genome assembly hg19 using TopHat v2.1.0 with fusion-search option based on Bowtie algorithm and fusion candidates were selected with the following requirements: each read mapping across a fusion point have ≥13 bases matching on both sides of the fusion, with no more than two mismatches; and the two ‘sides’ of the candidate transcripts either reside on different chromosomes or reside on the same chromosome and are separated by ≥100 kbp [64].

### 4.3. RT-qPCR

Gene expression levels were determined by RT-qPCR. cDNA was generated from total RNA using a reverse transcriptase Superscript III (Invitrogen, Carlsbad, CA, USA) according to the manufacturer’s instructions. RT-qPCR was performed using KAPA SYBR FAST qPCR kits (Kapa Biosystems, Inc., Woburn, MA, USA) and a StepOnePlus Real-Time PCR system (Thermo Fisher Scientific, Foster City, CA, USA) with the following steps: 95 °C for 20 s, followed by 40 cycles at 95 °C for 3 s, 60 °C for 30 s. The following primers were used for amplification: *SPON1-TRIM29* fusion mRNA, forward 5′-GACCATGCTGGGACCTTCC-3′, reverse 5′-CTTGATGCGGTCCTTCTCCTT-3′; and *GAPDH*, forward 5′-GGTGGTCTCCTCTGACTTCAACA-3′, reverse 5′-GTGGTCGTTGAGGGCAAATG-3′. The mRNA levels of the *SPON1-TRIM29* fusion transcript were normalized with *GAPDH*. The amplified products were subcloned into pBluescript-SK(-) (Stratagene, La Jolla, CA, USA) and sequenced using the Applied Biosystems 3500 Genetic Analyzer (Applied Biosystems, Foster City, CA, USA).

### 4.4. Cloning of SPON1-TRIM29 Fusion Transcript and Transfection

The *SPON1-TRIM29* fusion transcript was amplified from the patient’s cDNA using PCR with primers: forward, 5′-ATTGAATTCAGGCTGTCCCCGGCGCCCCTGAAG-3′ and reverse, 5′-ATTCTCGAGTCATGGGGCTTCGTTGGACCCAAT-3, and cloned into pcDNA3 (Invitrogen, Carlsbad, CA, USA) with C-terminal Flag-tag. 293T and A2780 cells were transfected with the SPON1-TRIM29 expressing vector or empty vector using FuGENE HD (Roche Diagnostics, Indianapolis, IN, USA) according to manufacturer’s instructions. For A2780 stable transfection, cells were selected with G418 and stable transformants (SPON1-TRIM29-A2780 #A and #B, and Vector-A2780 #A and #B) were established.

### 4.5. Small Interfering RNA

Small interfering RNAs (siRNAs) were designed and synthesized against the surrounding sequence of the identified *SPON1-TRIM29* joint point as follows (Sigma-Aldrich, St. Louis, MO, USA): siFusion #1, 5′-GGACCUUCCAGACGGAGCUdTdT-3′ (sense) and 5′-AGCUCCGUCUGGAAGGUCCdTdT-3′ (antisense); and siFusion #2, 5′-CCUUCCAGACGGAGCUGUCAUdTdT-3′ (sense) and 5′-AUGACAGCUCCGUCUGGAAGGdTdT-3′ (antisense). Two control siRNAs not targeting human transcripts (siLuc and siControl) were synthesized as follows (RNAi Inc, Tokyo, Japan): siLuc, 5′-GUGGAUUUCGAGUCGUCUUAA-3′ (sense) and 5′-AAGACGACUCGAAAUCCACAU-3′ (antisense); and siControl, 5′-GUACCGCACGUCAUUCGUAUC-3′ (sense) and 5′-UACGAAUGACGUGCGGUACGU-3′ (antisense).

### 4.6. Western Blot

Cell lysates were prepared with sample buffer consisting of 50 mM Tris-HCl (pH 6.8), 2% SDS, 10% glycerol, 1% 2-mercaptoethanol, and 0.01% BPB followed by boiling for 10 min. Equal amounts of the cellular proteins were separated by 10% SDS-PAGE and electrophoretically transferred onto PVDF membranes (Millipore, Billerica, MA, USA). The membranes were blocked with 5% skim milk in TBS-T (TBS containing 0.1% Tween-20) for 30 min and incubated with primary antibodies against Flag (1:3000; F3165, Sigma-Aldrich, St. Louis, MO, USA) or β-actin (1:4000; AC-74, Sigma-Aldrich, St. Louis, MO, USA) overnight at 4 °C. After washing with TBS-T, the membranes were then incubated with horseradish peroxidase-conjugated secondary antibody (anti-mouse immunoglobulin G (IgG), 1:4000; Thermo Fisher Scientific, Foster City, CA, USA) for 1 h at room temperature. Signals on the blots were detected with ECL Plus chemiluminescence detection kit (Thermo Fisher Scientific, Foster City, CA, USA).

### 4.7. Cell Proliferation Assay

Cells were seeded in 96 well-plate at a density of 5000 cells/well (Day 0) and cultured overnight. Next day (Day 1), Cisplatin, Paclitaxel, Olaparib, or vehicle was added into the media at the indicated concentrations. At Day 1, Day 3, and Day 5, the medium was removed and the plates were stored at −30 °C until DNA assay. Cell growth was evaluated by measuring cellular DNA content using bisbenzimidazole (Hoechst 33258, Invitrogen, Carlsbad, CA, USA) [6]. Briefly, the stored plates were again freeze and thawed with 100 μL/well of water and then added with 100 μL/well of TNE buffer (10 mM Tris-HCl [pH 7.5], 2 mM NaCl, and 1 mM EDTA) containing 20 μg/mL Hoechst 33258. Fluorescence in each well was measured on an ARVO5 multi-plate reader (Perkin Elmer, Waltham, MA, USA).

### 4.8. Cell Cycle Analysis

After treatment with cisplatin, paclitaxel, olaparib, or vehicle for 5 days at the indicated concentrations, cells were fixed with 70% ethanol. After fixing, cells were suspended in PBS, 0.8 mg/mL RNase A, 10 μg/mL of propidium iodide (Sigma, St. Louis, MO, USA) and incubated at room temperature for 30 min. The samples were sorted based on DNA content by using fluorescence-activated cell sorting (FACS) (FACScalibur; Becton Dickinson, Cockeysville, MD, USA) and the CellQuest software (Becton Dickinson, Cockeysville, MD, USA) in order to determine the percentages of cells that were in the G1, S, and G2/M phases of the cell cycle.

### 4.9. Apoptosis Assay

Cells were seeded in 6 cm-dish (3 × 10^4^ cells/dish) (Day 0). Next day (Day 1), 1 μM Cisplatin, 5 nM Paclitaxel, 2 μM Olaparib, or vehicle was added into the media. On Day 5, the cells were harvested, washed twice with ice-cold PBS, and dual stained using FITC Annexin V apoptosis Detection Kit I (BD Biosciences, Franklin Lakes, NJ, USA) according to the manufacturer’s protocol. Stained cells were analyzed using FACS Calibur Flow Cytometer (Becton Dickinson, Cockeysville, MD, USA).

### 4.10. Tumor Growth in Athymic Mice

All animal experiments were approved by the Animal Care and Use Committee of Saitama Medical University and conducted in accordance with the Guidelines and Regulations for the Care and Use of Experimental Animals by Saitama Medical University. SPON1-TRIM29-A2780 #A and Vector-A2780 #A cells (5 × 10^6^ cells per 0.15 mL phosphate-buffered saline) suspended in Matrigel (BD Biosciences, San Jose, CA, USA) were injected subcutaneously into female athymic mice (8 week-old BALB/c nu/nu) (Day 0). Tumor volumes were calculated at Day 0, 2, 4, and 9 by measuring the tumor radii. From the next day after the tumor inoculation, mice were treated with 5 μg siFusion #1 or siControl every 3rd day. Fifty microliters of a solution containing the siRNAs, 4 μL of GeneSilencer Reagent (Gene Therapy Systems, La Jolla, CA, USA), and DMEM was injected into the tumor.

## Figures and Tables

**Figure 1 ijms-23-00689-f001:**
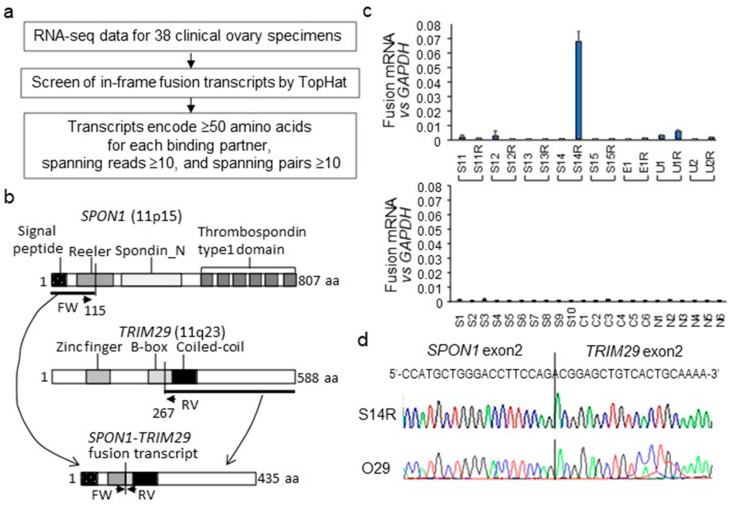
Identification of *SPON1-TRIM29* fusion transcript in ovarian cancer by RNA-seq. (**a**) Schematical presentation of strategies to identify fusion transcripts in ovarian cancers by RNA-seq. (**b**) Schematic presentation of the putative SPON1-TRIM29 fusion protein translated from the *SPON1-TRIM29* fusion transcript in which the *SPON1* exon2 is fused in-frame to the *TRIM29* exon2. SPON1 and TRIM29 proteins are represented with their domain structures. Primers for specific amplification of *SPON1-TRIM29* fusion mRNA in RT-qPCR are indicated as arrows (FW and RV). (**c**) Expression levels of *SPON1-TRIM29* fusion transcript in 38 clinical ovary specimens analyzed by RT-qPCR. cDNA samples prepared from clinical specimens were amplified using primer sets that span the fusion point. Upper panel for 8 matched pairs of primary and recurrent ovarian cancer specimens. Lower panel for 10 high-grade serous carcinomas (HGSC, #S1–S10), 6 clear cell carcinomas (CCC, #C1–#C6), and 6 normal ovary tissues (#N1-N6). The mRNA levels of *SPON1-TRIM29* fusion transcript are normalized with *GAPDH* and the data are presented as means ± SD (*n* = 3). (**d**) Chromatograms for Sanger sequencing results for *SPON1-TRIM29* fusion transcript in tumors #S14R and #O29.

**Figure 2 ijms-23-00689-f002:**
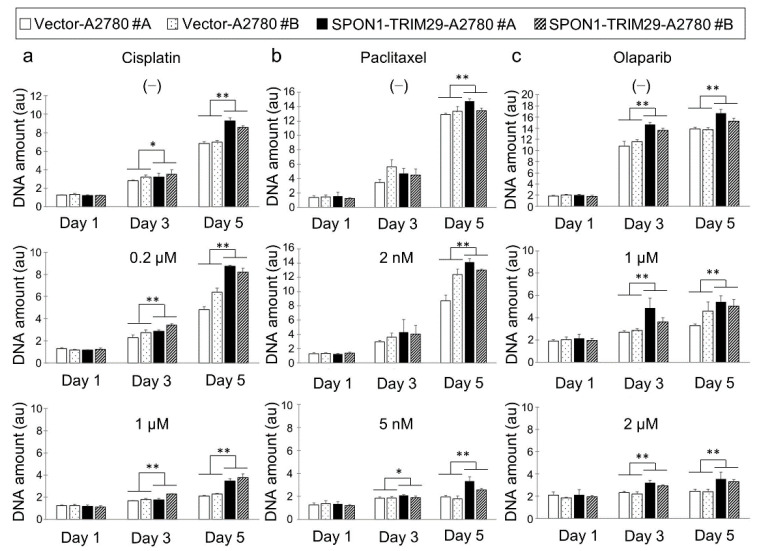
SPON1-TRIM29 expression blunts chemotherapy drug sensitivity in ovarian cancer cells. (**a**–**c**) *SPON1-TRIM29*- and vector-transfected A2780 cells were seeded in 96 well-plate on day 0 and treated with either cisplatin (**a**), paclitaxel (**b**), or olaparib (**c**) at the indicated concentrations on day 1. Vehicle treatment corresponding to each drug treatment is indicated as (−). The medium was removed at the indicated time points and DNA assay was performed to estimate cell proliferation. Data are presented as means ± SD (*n* = 3). * *p* < 0.05, ** *p* < 0.01, two-way ANOVA.

**Figure 3 ijms-23-00689-f003:**
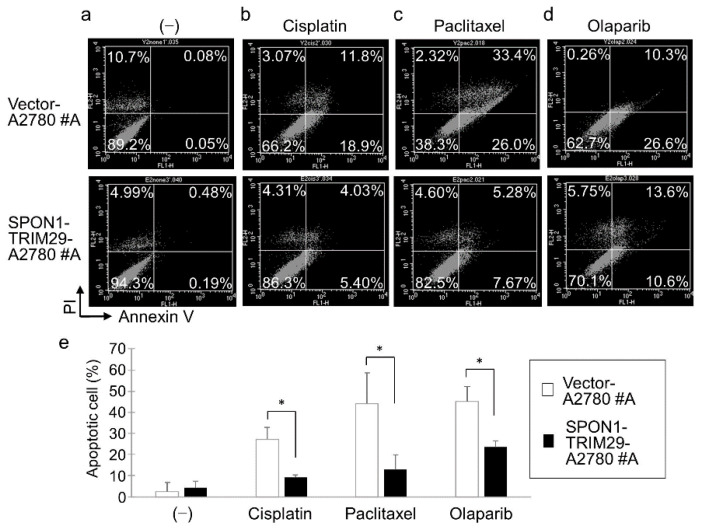
SPON1-TRIM29 expression inhibits cancer drug-induced apoptosis in ovarian cancer cells. (**a**–**d**) Vector-A2780 #A and SPON1-TRIM29-A2780 #A cells were seeded in 6 cm-dish on day 0. Either vehicle (−) (**a**), 1 μM cisplatin (**b**), 5 nM paclitaxel (**c**), or 2 μM olaparib (**d**) was added into the media on day 1 and the culture was continued. On day 5, cells were harvested and stained with annexin V and propidium iodide (PI). (**e**) Apoptosis was measured using flow cytometry and percentages of annexin V-positive apoptotic cell population were determined. Data are presented as means ± SD (*n* = 3). * *p* < 0.05, Student’s *t*-test.

**Figure 4 ijms-23-00689-f004:**
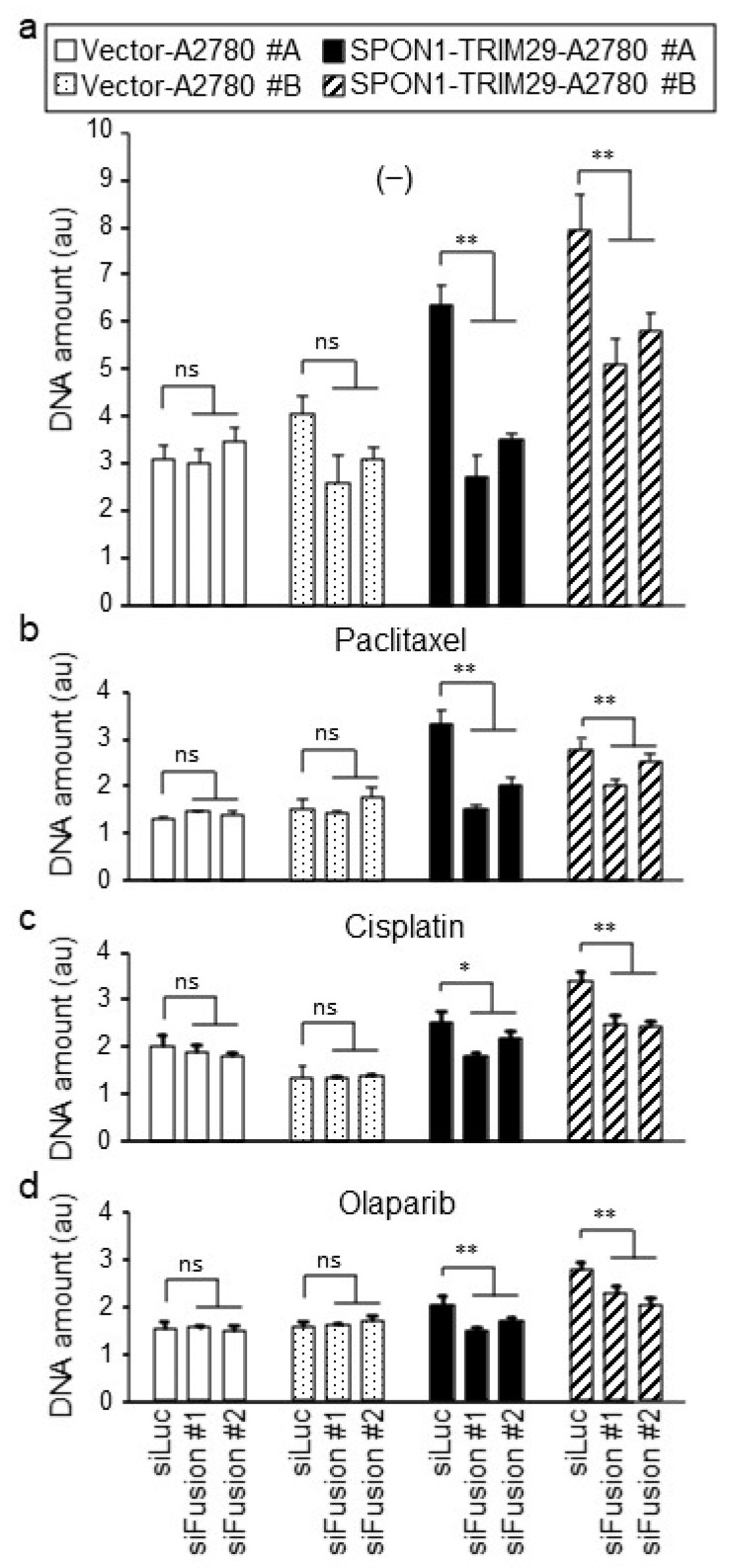
SPON1-TRIM29 blockade impairs cancer drug-resistant growth in SPON1-TRIM29-A2780 cells. SPON1-TRIM29- and Vector-A2780 cells were treated with siRNAs specific for *SPON1-TRIM29* joint sequence (siFusion #1 and #2) or control siRNA (siLuc), and treated with vehicle (−) (**a**), 1 μM cisplatin (**b**), 5 nM paclitaxel (**c**), or 2 μM olaparib (**d**). Cell growth was estimated by DNA assay. Data are presented as means ± SD (*n* = 5). * *p* < 0.05; ** *p* < 0.01, Student’s *t*-test. ns, not significant.

**Figure 5 ijms-23-00689-f005:**
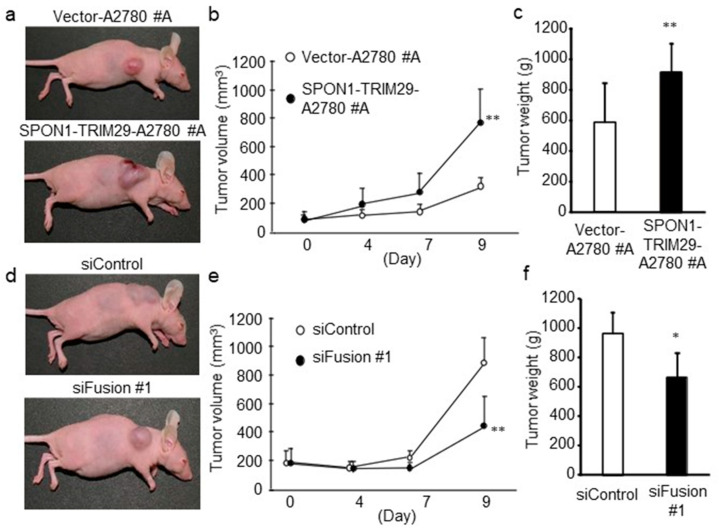
SPON1-TRIM29 expression promotes tumor growth of A2780 cells. (**a**–**c**) Increased tumor growth of SPON1-TRIM29-A2780 cells in athymic mice. Female athymic mice were inoculated with SPON1-TRIM29- or Vector-A2780 cells and the representative tumor-bearing mice at end point are indicated (**a**). Tumor volumes (mm^3^) were estimated at the indicated time points (**b**) and dissected tumor weights at end point were indicated (**c**). (**d**–**f**) siRNA targeting *SPON1-TRIM29* decreases tumor growth of SPON1-TRIM29-A2780 cells in athymic mice. Female athymic mice were inoculated with SPON1-TRIM29-A2780 cells and then administrated with control siRNA (siControl) or *SPON1-TRIM29* siRNA (siFusion #1) every 3 days and the representative tumor-bearing mice at end point are indicated (**d**). Tumor volumes (mm^3^) were estimated at the indicated time points (**e**) and dissected tumor weights at end point were indicated (**f**). Data are presented as means ± SD (*n* = 7). * *p* < 0.05, ** *p* < 0.01, Student’s *t*-test.

**Table 1 ijms-23-00689-t001:** Clinicopathological characteristics of 32 ovarian cancer tissues from 24 patients and 6 normal tissues.

Characteristics	Ovarian Cancer Subtype or Normal Tissue				
	High-Grade Serous	Clear Cell	Endometrioid	Unclassified	Normal Tissue ^a^
Total cases ^b^	20 (0)	6 (0)	2 (0)	4 (0)	6 (0)
Primary	15 (0)	6 (0)	1 (0)	2 (0)	
Recurrent ^c^	5 (1)	0 (0)	1 (0)	2 (0)	
Patient age ^d^					
Average	61.5	57.5	57.0	74.0	53.7
Median	63.0	59.5	57.0	74.0	53.0
Range	40−80	50−63	57−57	72−76	41−66
FIGO Stage ^e^					
I, II	0	2	1	0	
III, IV	15	4	0	2	

^a^ Among 6 normal tissues, 5 tissues were obtained from scrape biopsy of the contralateral unaffected ovary in patients with unilateral ovary tumor and one normal tissue was derived from ipsilateral oviduct of benign ovarian tumor (mucinous cyst adenoma); ^b^ The number of *SPON1-TRIM29* fusion detected cases by RNA-seq was indicated in parentheses; ^c^ Recurrent tumors matched with the corresponding primary tumors; ^d^ Ages at diagnosis were used for statistical analysis; ^e^ FIGO stage was available for primary tumors.

## Data Availability

The data used to support the findings of the present study are available from the corresponding author upon request.

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
