# Peer review of "Identification of a Novel Oncogenic Fusion Gene SPON1-TRIM29 in Clinical Ovarian Cancer That Promotes Cell and Tumor Growth and Enhances Chemoresistance in A2780 Cells"

_ijms, 2022, doi:10.3390/ijms23020689_

Round 1

Reviewer 1 Report

In this paper, the authors identified a novel oncogenic fusion gene in ovarian cancer tumor tissues and explored its effect on the proliferation and drug resistance of A2780 cell line in vitro and in vivo. However, the following issues should be modified to be accepted for publishing.

1、Fig.4: Although the amount of SPON1-TRIM29 expression in siFusion transfected SPON1-TRIM29-A2780 cells was significantly decreased compared with untransfected control group, there was still a small amount of SPON1-TRIM29 expression. Whereas SPON1-TRIM29 is not expressed in vector-A2780. The experimental results show that siFusion group is more easily inhibited by chemotherapeutic drugs, in contrast to the article's conclusions. Please confirm the experimental results.

2、This study verified the effect of SPON1-TRIM29 fusion gene on the proliferation and drug resistance of A2780 cell line in vitro and in vivo, but could not fully reflect the effect of SPON1-TRIM29 fusion gene on ovarian cancer. It is recommended to increase the number of related experiments, increase the number of ovarian cancer cell lines, or use primary ovarian cancer cells for experiment.

3、Apoptosis experiment: cells not treated with chemotherapy drugs die by more than 10%, are there poor cell states or other problems? It is recommended to repeat to increase the results credibility.

4、Figure 5A C is suggested to be drawn as a broken line diagram.

5、Figure 5: photographs of all nude mice as well as pictures of ex vivo tumors with a ruler are required to be added to figure 5. Figure formats refer to PMID: 32019566.

Reviewer 2 Report

Summary: The manuscript reports the identification of an in-frame fusion transcript SPON1-TRIM29 in chromosome 11 from a high-grade serous carcinoma (HGSC) specimen and its expression in another tumor from a distinct HGSC patient. Its stable expression in ovarian cancer A2780 cells increases cell growth but decreases apoptosis even if anticancer drugs like cisplatin, paclitaxel, or olaparib are administered.

Strengths: The study presents an interesting finding on fusion transcript expression in cancer which not only reduces apoptosis under anti-cancer drug treatment but also increases cell growth.

Weaknesses: 1. It is not clearly mentioned exactly how many individual patients showed the expression of the fusion transcript, for each kind of cancers. A small table can be included in figure 1 to clarify that information.

2. What do the abbreviations in X-axis of figure 1 C top panel mean?

Based on the expression patterns in Figure 1 C top & bottom panels, which samples are considered to be expressing the fusion transcript? S14R has a very detectable level but are any of the others being considered here? If yes, what is the basis for cut-off?

3. At least 1 representative raw image file for each of the 3 drugs and the control which were used to quantify the cell proliferation and apoptosis, need to be included.

4. In section 2.3, it needs to be included that annexin-V detects apoptotic cells because it binds to an apoptotic marker on the cell membrane. That will clarify to the reader about the correlation between annexin-V and aopotosis.

5. Page 10 lines 319-321 : Any perspectives will be helpful.

Round 2

Reviewer 1 Report

In this paper, the authors have carefully revised the article, however, there are still some issues that need to be revised or explained before publication.

  1. Figure 4B: The author explained the experimental process in detail and modified some pictures, but still did not explain the previous problems well. The author only made possible assumptions based on the experimental results, which could not support the conclusion of the article.

  1. Part of the references are old, it is recommended to cite the latest research progress.

Reviewer 2 Report

In the revised version, the comments have been addressed and manuscript is improved.

Author Response

We appreciate the Reviewer's comment.

Round 3

Reviewer 1 Report

In this paper, the authors identified a novel oncogenic fusion gene that promotes proliferation of ovarian cancer cells, and performed functional studies of this fusion gene within the ovarian cancer A2780 cell line and in nude mice. With modifications, the authors have solved most of the problems. However, before publication, we still have some suggestions to hope the authors to adopt.

  1. This study was mainly conducted using the ovarian cancer A2780 cell line as a carrier, which the authors are advised to state in the title.

  1. The SPON1-TRIM29 fusion was first detected in a recurrent HGSC tumor whereas not in the corresponding primary cancer. It is recommended that authors state this in the abstract.
